# Investigating the Impact of Topology and Physical Impairments on the Capacity of an Optical Backbone Network

**Alexandre Freitas [1] and João Pires [2],***

[1] Department of Electrical and Computer Engineering, Instituto Superior Técnico, Universidade de Lisboa, Avenida Rovisco Pais 1, 1049-001 Lisboa, Portugal; alexandre.costa.freitas@tecnico.ulisboa.pt
[2] Department of Electrical and Computer Engineering and Instituto de Telecomunicações, Instituto Superior Técnico, Universidade de Lisboa, Avenida Rovisco Pais 1, 1049-001 Lisboa, Portugal
\* Correspondence: jpires@lx.it.pt

**Abstract:** Optical backbone networks constitute the fundamental infrastructure employed today by network operators to deliver services to users. As network capacity is a key factor influencing optical network performance, it is important to understand how topological and physical properties impact its behavior and to have the capability to estimate its value. In this context, we propose here a method to evaluate the network capacity that relies on the optical reach to account for physical layer aspects in conjunction with constrained routing techniques for traffic routing. As this type of routing can lead to traffic blocking, particularly due to the limitation on the number of wavelengths per fiber, we also propose a fiber assignment algorithm designed to deal with this problem. We apply this method to a set of randomly generated networks using a modified Waxman model, and for a network with 60 nodes, in a scenario without blocking, we obtain capacities of about 2.5 Pbit/s for a symbol rate of 64 Gbaud and about 5 Pbit/s for a symbol rate of 128 Gbaud. Remarkably, this duplication in the total network capacity is achieved by an increase in the total fiber length of only about 51%.

**Keywords:** network capacity; fiber assignment; random networks; optical networks; optical communications

## 1. Introduction

In recent years, there has been an enormous growth in telecommunications traffic due to the surge of applications and services that require high bandwidth and generate large amounts of data, such as video streaming services, social media platforms, cloud computing, and the adoption of emerging technologies such as 5G, artificial intelligence, etc. This evolving landscape requires the use of very high-speed telecommunications networks like optical networks [1].

Optical networks are communication infrastructures that utilize light for transmission, processing, and routing information and rely on optical fibers as their transmission medium. These networks vary in terms of distance and capacity, falling into several tiers: (1) Backbone networks, which span extensive geographic distances and offer huge capacities (in the order of dozens of Tbit/s); (2) Metro networks, which cover cities or metropolitan areas, handling data transmission in the range of hundreds of Gbit/s; (3) Access networks, also known as 'last-mile networks', which encompass small areas, connecting end-users to the network providers and delivering data rates on the order of a few Gbit/s.

Wavelength Division Multiplexing (WDM) is a fundamental technology in the optical networking field, as it enables the transmission of large amounts of data across long distances. It works by simultaneously transmitting multiple optical signals, often referred to as optical channels, through a single optical fiber, with each channel utilizing its own wavelength. The number of channels transmitted per fiber depends on both the spacing between wavelengths and the WDM signal bandwidth, which in turn is limited by the bandwidth of the optical amplifiers used to compensate for the fiber losses. The most

commonly used optical amplifier is the Erbium-Doped Fiber Amplifier (EDFA), which, utilizing standard technology, can provide an amplification bandwidth of approximately 4800 GHz, although more advanced solutions can achieve values up to 6000 GHz. Therefore, for a typical channel spacing of 50 GHz, the first solution can support up to 96 channels, while the second one can accommodate up to 120 channels [2].

Network capacity is an important performance feature of optical networks. This capacity can be defined as the maximum amount of data that the entire network can handle per unit of time, and it is closely related to channel capacity. The concept of channel capacity was introduced by Claude Shannon in 1948 [3]. This refers to the maximum data rate at which the information can be reliably transmitted through a noisy channel without errors. The fundamental assumptions behind this definition are that the noise is additive, white, and Gaussian (AWGN) and that the channel is linear, i.e., the capacity always increases with increasing signal power. However, the last assumption does not hold for optical fiber channels, which are nonlinear by nature. This behavior implies that the optical channel capacity does not grow indefinitely; instead, it is limited and reaches a maximum value as the transmitted signal power increases [4–6].

When estimating the capacity of an optical network, one must necessarily consider the optical channel capacity. However, the problem is more complex than that, as it is necessary to also consider topological aspects, traffic demands, routing, and wavelength and modulation assignments. In other words, this capacity estimation can be viewed as a multilayer problem in the sense that it requires taking into account not only physical layer properties but also network layer aspects. Furthermore, for an optimized design, it would be paramount to have a clear understanding of how these different aspects correlate with the network capacity. For that purpose, it is convenient to have available a large number of network topologies, which can be obtained using, for example, generative graph models [7].

The problem of estimating the optical channel capacity has been the focus of many studies. Some rely on accurate numerical simulations [6], while others offer detailed analytical models based on either the Gaussian noise (GN) model [8–10] or a regular perturbation model [11]. More recently, the topic of optical network capacity has also received some attention. In [12], the authors presented an algorithm to maximize the capacity of an optical network in the presence of physical layer impairments. The algorithm was based on an integer linear program (ILP) and was designed with the goal of optimizing routing, wavelength assignment, modulation format, and launched power allocation. An alternative approach for capacity estimation using a heuristic algorithm for routing and wavelength assignment instead of the ILP was provided in [13]. To understand how network topology characteristics influence network capacity , a new generative graph model was developed [14]. This model is based on the classical Barabási-Albert model, which has been properly modified to incorporate physical layer aspects. The published results showed that it can maximize the network capacity in comparison with classical models. Recently, a framework was also proposed to study the relationship between various topological parameters and network performance metrics, including network capacity [15]. That framework provided valuable insights into the key parameters that affect network capacity.

Apart from the last work, which relies on dynamic routing, all the other referred studies on network capacity used static routing with no channel blocking. However, since the number of optical channels per optical fiber is limited, it makes sense to also use a constrained routing approach, as this limitation can lead to blocking under certain conditions. Another topic that deserves consideration is studying how the symbol rate (also referred to as baud rate) impacts network capacity. In fact, considerable research has been conducted to increase the symbol rate within optical networks. Currently, commercial deployments typically operate between 60–90 Gbaud, while field trials have reached 130 Gbaud [16], and laboratory demonstrations have achieved symbol rates of 200 Gbaud [17].

This paper is focused on the topic of capacity in optical backbone networks and examines how different network and physical layer parameters influence its value, giving special emphasis to the symbol rate. We present an alternative approach to evaluating the capacity of optical networks that uses a constrained routing algorithm to account for the limitations in the number of optical channels and use the metric optical reach, which measures the maximum distance an optical channel can effectively propagate, to describe the impact of the physical layer. Furthermore, a strategy to address the blocking caused by insufficient spectral resources (wavelengths) by adding additional optical fibers is also proposed. The paper's results are obtained across hundreds of network topologies generated using the modified Waxman method.

The rest of the paper is organized as follows: Section 2 reviews the concept of channel capacity and introduces the necessary background to determine optical reach. Section 3 defines the method used to generate random networks. Section 4 introduces a suitable approach to computing the optical network capacity, taking into account the constraints due to the limited number of optical channels per fiber, while Section 5 describes a strategy to overcome blocking by adding more fibers per link. Section 6 provides some simulation results, and finally, Section 7 summarizes and concludes the paper.

## 2. Optical Channel Capacity

An optical channel can be seen as a communication pathway through which information is transmitted in the optical domain from a sender to a receiver, utilizing an optical fiber as a transmission medium. This channel is characterized by its carrier frequency, denoted as $\nu_c$ (or carrier wavelength $\lambda_c$) and occupied bandwidth, denoted as $B_{ch}$. The minimum bandwidth that guarantees a signal transmission over the channel without inter-symbol interference is defined using the Nyquist criterion and is equal to the symbol rate $R_s$ [4]. The capacity of an optical channel is defined as the maximum data rate at which the information can be effectively transmitted through the channel. This capacity is typically expressed in bit/s. This capacity can be calculated using Shannon's theory [3] under the assumption that the noise sources present in these channels are modeled as AWGN sources, giving [6]

$$C_{ch} = 2R_s\log_2\left(1 + SNR\right) \qquad [\text{bit/s}] \tag{1}$$

where $SNR$ is the signal-to-noise ratio at the receiver side computed for a channel bandwidth equal to $R_s$, given by

$$SNR = \frac{P_{ch}}{N_0 R_s} \tag{2}$$

where $P_{ch}$ is the average optical power per channel in watts, and $N_0$ is the noise power spectral density (PSD) in watt/Hz. Note that factor 2 in (1) stems from the fact that the optical fiber channel supports two optical channels with orthogonal polarizations, commonly referred to as polarization multiplexed (PM) optical channels.

One important noise source in optical communications systems is the amplified spontaneous emission (ASE) noise. This noise is generated inside optical amplifiers simultaneously with signal amplification and can be effectively described using a random optical field with statistical properties like those of AWGN noise [6]. Optical amplifiers are used to compensate for the optical fiber losses. To achieve this, optical amplifiers, typically EDFAs, are placed at discrete intervals along an optical link, with each amplifier exactly compensating for the loss incurred by each fiber span. For a link of length $L$ composed of $N_s$ identical spans, the span length is $L_s = L/N_s$, while the span attenuation is $A_s = \alpha L_s$, where $\alpha$ is the fiber attenuation coefficient in dB/km. Typically, $\alpha$ is approximately 0.2 dB/km within the 1550 nm wavelength region, denoted as C-band. The PSD of the ASE noise at the end of a chain of $N_s$ amplifiers, spaced by fiber spans of length $L_s$, is given by

$$N_{ase} = N_{ase,1}N_s = h\nu_c f_n(a_s - 1)N_s \tag{3}$$

where $N_{ase,1}$ is the ASE per span, $h$ is the Planck's constant (in joule-second), $f_n$ is the noise figure ($f_n = 10^{F_n/10}$, with $F_n$ in dB), and $a_s = 10^{A_s/10}$.

Another significant noise source is nonlinear interference (NLI), which results from the Kerr effect in optical fibers. The Kerr effect refers to the dependence of the refractive index of the fiber on the transmitted signal power. This characteristic makes the optical fiber channel intrinsically nonlinear and, in this sense, different from other transmission media used for information transfer that have a linear behavior. Interestingly, it has been demonstrated in [18] through simulations and experiments that the impact of NLI noise on WDM links, supported in dispersion uncompensated fibers, can also be modeled as additive Gaussian noise. Furthermore, it was shown in [9] that under specific conditions, such as the Nyquist limit, the white noise assumption leads to quite accurate results. Note that such a limit is achieved when all the WDM channels have a rectangular spectral width and a frequency spacing equal to $R_s$. This permits us to characterize the NLI noise also as an AWGN process with a power spectral density of $N_{nli}$. As the ASE and NLI noises are assumed to be uncorrelated, their power spectral densities simply add, resulting in $N_0 = N_{ase} + N_{nli}$. In these circumstances, the signal-to-noise ratio of an optical channel can be described as

$$SNR = \frac{P_{ch}}{(N_{ase} + N_{nli})R_s} \tag{4}$$

where $P_{ch}$ denotes the launched average optical power per channel.

Rigorous characterization of $N_{nli}$ is not an easy task, and many studies have been published on this topic (see, for example, [18,19]). Fortunately, some closed-form approximations have also been published [8,18], which simplifies the evaluation of $N_{nli}$. One of these approximations, which is based on the white noise assumption, allows to write the PSD of the NLI at the end of a fiber link with $N_s$ spans in the following way:

$$N_{nli} = \mu_n N_s P_{ch}^3 \tag{5}$$

where $\mu_n$ is the NLI coefficient per span given by

$$\mu_n \approx \acute{\mu}_n \frac{1}{R_s^{\,3}} = \left(\frac{2}{3}\right)^3 \gamma^2 L_{ef} \frac{\ln\left(\pi^2 |\beta_2| L_{ef} B_{WDM}^2\right)}{\pi |\beta_2|} \frac{1}{R_s^{\,3}}. \tag{6}$$

In the last equation, one can identify parameters related to the optical fiber, such as $\gamma$, the fiber nonlinear coefficient in $W^{-1}km^{-1}$, $\beta_2$, the fiber dispersion in $ps^2km^{-1}$ and $L_{ef}$, the span effective length in km. Additionally, there are parameters related to the signal, such as $B_{WDM}$, the optical bandwidth of the WDM signal in Hz, assumed to be composed of $N_{ch}$ channels spaced by $\Delta\nu_{ch}$, in such a way that $B_{WDM} = N_{ch}\Delta\nu_{ch}$. In addition, the span effective length is given as

$$L_{ef} = (1 - \exp(-2a_N L_s))/(2\alpha_N) \tag{7}$$

where $L_s$ is the span length and $a_N$ is the fiber attenuation coefficient in Np/km, i.e., $\alpha_N = \alpha_{dB/km}/20\log_{10} e$. Using (4)–(6), one arrives to

$$SNR = \frac{P_{ch}}{\left(N_{ase,1} + \mu_n P_{ch}^3\right)N_s R_s}. \tag{8}$$

From (8) one can derive the following equation for the optimum launch power [9]

$$P_{ch}^{opt} = R_s \sqrt[3]{\frac{N_{ase,1}}{2\acute{\mu}_n}}. \tag{9}$$

The maximum channel capacity can be determined by inserting (8) and (9), into (1), giving

$$C_{ch} = 2R_s \log_2 \left( 1 + \frac{L_s}{3L} \sqrt[3]{\frac{4}{\hat{\mu}_n N_{ase,1}^2}} \right).$$ (10)

From (9) and (10) we can see that:

(1) $P_{ch}^{opt}$ depends on ASE and NLI noise and varies linearly with the symbol rate. For the parameters given in Table 1 we arrive to $P_{ch}^{opt} = 0.89$ dBm for $R_s = 64$ Gbaud, and $P_{ch}^{opt} = 3.89$ dBm for $R_s = 128$ Gbaud.

(2) The channel capacity increases linearly with the symbol rate and decreases linearly with the total link length.

**Table 1.** Optical fiber and system parameters.

| Parameter | Symbol | Value |
|---|---|---|
| Fiber Attenuation Coefficient | $\alpha$ | 0.22 dB/km |
| Fiber Dispersion Parameter | $\beta_2$ | $-21.7$ ps$^2$km$^{-1}$ |
| Fiber Nonlinear Coefficient | $\gamma$ | 1.27 W$^{-1}$km$^{-1}$ |
| Carrier Frequency | $\nu_c$ | 193.41 THz |
| Carrier Wavelength | $\lambda_c$ | 1550 nm |
| Span length | $L_s$ | 80 km |
| EDFA noise figure | $F_n$ | d dB |
| Symbol rate | $R_s$ | 64 Gbaud, 128 Gbaud |
| Channel Spacing | $\Delta\nu_{ch}$ | 64 GHz, 128 GHz |
| Number of Channels | $N_{ch}$ | 75, 37 |
| WDM bandwidth | $B_{WDM}$ | 4800 THz |

The optical reach, also denoted as transmission reach, is an important parameter used in the context of this work to describe the impact of the physical layer on the performance of an optical channel. The optical reach is defined here as the maximum length of an optical channel for which a certain value of the capacity can be met. As this length can be viewed as the total link length $L$, one can use (10) to obtain the optical reach for various capacity values. Assuming, as seen before, that for the 64 Gbaud case, $L$ is a multiple of the span length, Table 2 shows the optical reach obtained using (10) for different values of the Shannon channel capacities. Furthermore, in the case of 128 Gbaud, we considered a 10% reach reduction compared to the previous scenario to address additional limitations not taken into account in the formulation that leads to (10) (see [19]). Although these capacity values can be seen as upper bounds, it is worth noting that a recent field trial reported an 800 Gb/s transmission over a distance of 6600 km for a symbol rate of 120 Gbaud [20], which is not far from the values of the reach given in Table 2 for that bit rate.

**Table 2.** Optical reach values for two symbol rates.

| Reach (km) 64 Gbaud | Capacity (Gb/s) 64 Gbaud | Reach (km) 128 Gbaud | Capacity (Gb/s) 128 Gbaud |
|---|---|---|---|
| 23,120 | 200 | 20,808 | 400 |
| 11,120 | 300 | 10,008 | 600 |
| 5840 | 400 | 5256 | 800 |
| 3280 | 500 | 2952 | 1000 |
| 1760 | 600 | 1584 | 1200 |
| 1040 | 700 | 936 | 1400 |
| 560 | 800 | 504 | 1600 |
| 320 | 900 | 288 | 1800 |
| 160 | 1000 | 144 | 2000 |
| 80 | 1100 | 72 | 2200 |

### 3. Network Topology Model

In an abstract way, an optical network can be described as an undirected graph $G(V, E)$, with $V = \{v_1, \ldots, v_N\}$ denoting a set of nodes and $E = \{e_1, \ldots, e_K\}$ denoting a set of links, where $N = |V|$ is the number of nodes, and $K = |E|$ is the number of links. In transparent optical networks, all node functionalities take place in the optical domain, and the nodes are built upon reconfigurable optical add-drop multiplexers (ROADMs). Meanwhile, an optical link represents a physical interconnection between two nodes, implemented using optical fibers and optical amplifiers. In bidirectional links, some fibers are used in one direction and others (typically the same number) in the opposite direction. Each optical fiber supports WDM signals, meaning it carries a specific number of optical channels.

Besides $N$ and $K$, other important parameters are the node degree $\delta(G)$, the network diameter $diam(G)$, and the edge connectivity $\lambda(G)$. $\delta(G)$ defines the number of links connected to a node, $diam(G)$ is the length of the longest shortest path between any two nodes, while $\lambda(G)$ represents the maximum number of link-disjoint paths between two nodes. The $\lambda$-connectivity is a measure of a network's resilience against link failures, making it a key parameter in designing protection paths in optical networks.

To have a clear understanding of how different topological parameters impact network capacity, it is paramount to have large numbers of network topologies available, which can be obtained from a set of random graphs designed to adequately describe the characteristics of real-world optical networks. Erdős–Rényi and Waxman models are widely used to generate random networks. The last model works by randomly placing nodes in a two-dimensional space with specific coordinates and connecting them with links based on a probability function determined using the distance between those nodes. In the Waxman model, the probability that node $i$ establishes a link to node $j$ is given by [7]:

$$P(i, j) = \beta \exp \frac{-d(i, j)}{L_w \alpha} \tag{11}$$

where $d(i, j)$ is the Euclidean distance between the nodes, $L_w$ is the maximum distance between any two nodes, and $\alpha$ and $\beta$ are parameters in the range of 0 to 1.

In contrast, the first model does not reference node positions, and the links are added with a uniform probability. Assigning the nodes' positions in space makes the Waxman model better suited for describing realistic optical networks. However, the Waxman model cannot generate $\lambda$-connected graphs, which is a significant limitation in the context of optical backbone networks, where survivability is a primordial feature. To overcome such a limitation, one uses the modified Waxman model [7] in this work.

The modified Waxman model is designed to generate optical backbone networks that are survivable to single-link failures and are conceived as interconnected sets of subnetworks. In this sense, the two-dimensional space is divided into a set of regions where nodes are randomly placed in the first part of the process. In the subsequent steps, nodes are interconnected within each region and then across different regions according to the Waxman probability, subject to certain constraints in terms of node degree and $\lambda$-connectivity. For exemplification purposes, Figure 1 shows a generated graph with $N = 10$, $K = 20$, which gives an average node degree of $<\delta> = 2K/N = 4$. Two distinct subnetworks $S_1$ and $S_2$ are clearly identified within the graph, with $S_1 = \{0, 1, 2, 3\}$ and $S_2 = \{5, 7, 8, 9\}$, corresponding to the aforementioned regions. Furthermore, $\lambda(G) = 3$, with this value being determined by calculating the minimum number of links that need to be removed to disconnect the graph.

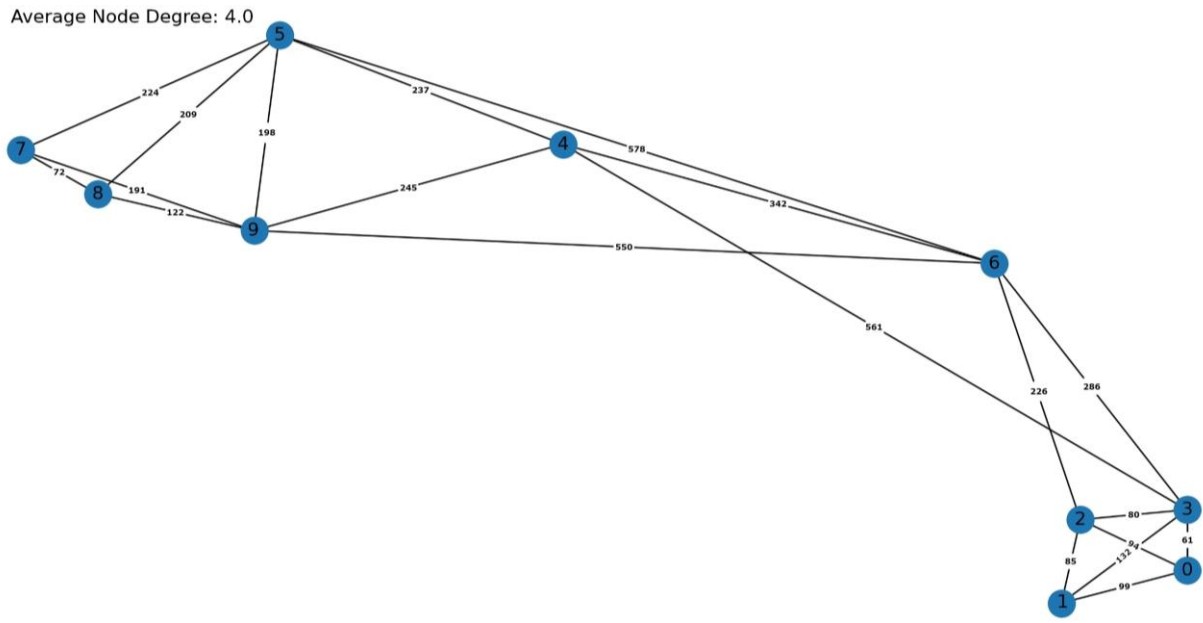

**Figure 1.** Network topology generated using the modified Waxman model with $N = 10$, $K = 20$, and $\lambda(G) = 3$. The regions $S_1 = \{0, 1, 2, 3\}$ and $S_2 = \{5, 7, 8, 9\}$ can be identified.

To ensure that the generated graphs accurately mimic real optical backbone networks, it is important to compare certain statistics. In [7], it is demonstrated that the node degree distribution of these networks follows a Poisson distribution. Figure 2 shows that the node degrees of the random graphs generated using the modified Waxman model closely approximate the Poisson statistics seen in real networks.

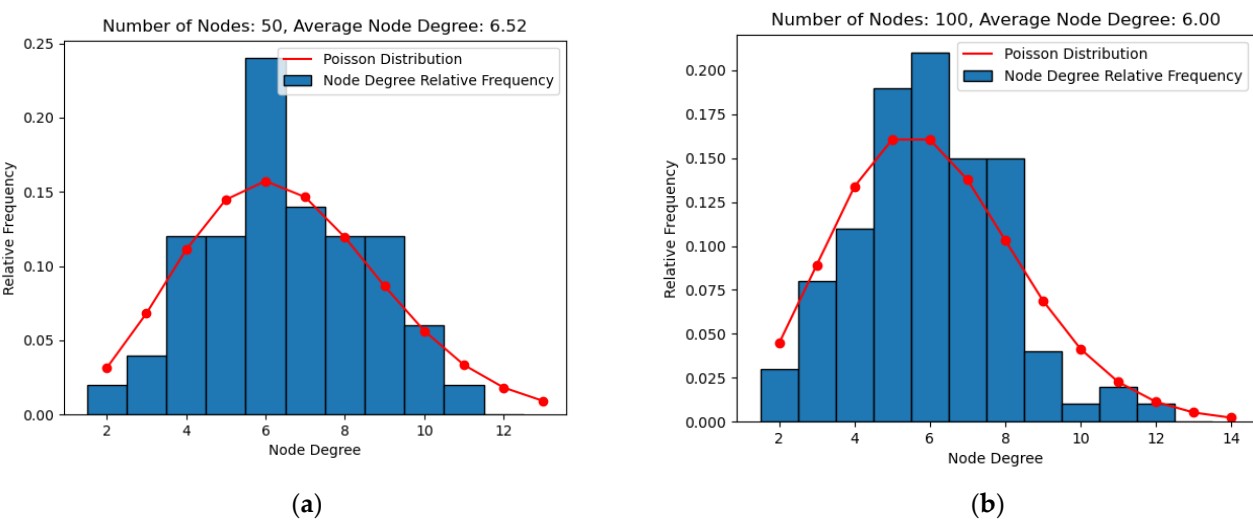

**Figure 2.** Average node degree frequency and Poisson distribution for random graphs generated with the modified Waxman model: (**a**) $N = 50$ and $< \delta > = 6.52$; (**b**) $N = 100$ and $< \delta > = 6.0$.

## 4. Constrained Routing and Network Capacity

Network capacity, also known as throughput, can be defined as the maximum amount of data that a network can handle per unit of time. This capacity depends on various network properties such as the physical and logical topology (traffic profile), optical reach, link capacity, node structure, routing, wavelength assignment, etc. Physical topology describes the interconnection pattern of nodes and is typically known in advance. Nodes are considered simultaneously as the source and destination of traffic. A starting point in the network capacity evaluation is the definition of the traffic demand profile. This profile

is defined by the traffic matrix $T = [t_{s,d}]$, where each entry $t_{s,d}$ represents a traffic demand, or in other terms, the volume of traffic flowing from a source node $s$ to a destination node $d$, with $s, d \in V$. In this analysis, it is assumed that the traffic profile is uniform and equal among all node pairs, which corresponds to

$$t_{s,d} = \begin{cases} 1 & s \neq d \\ 0 & s = d \end{cases} \tag{12}$$

Note that this traffic profile describes a full-mesh logical topology in the sense that each node is logically connected to every other node within the network [21]. Another important point in the network capacity evaluation is the link characterization. The link $(i, j) \in E$ can be described by two attributes: (1) length $l_{i,j}$; (2) capacity $c_{i,j}$ determined by the number of optical channels $N_{ch}$ available in the links given by $c_{i,j} = N_{ch}$. As already seen, this number is limited by the bandwidth $B_{WDM}$ and the symbol rate $R_s$.

For each traffic demand, it is necessary to find a path in the physical topology between each pair of nodes. This process is known as routing. Since there are multiple paths between each pair of nodes, the objective is to determine the shortest path using a heuristic like Dijkstra's algorithm. The shortest path corresponds to the one that minimizes the total path length, defined as the sum of the lengths of all the links traversed by the path. However, in this case, the routing is constrained by the capacity $c_{i,j}$ leading to the concept of constrained routing (CR) problem [22]. The objective of this problem is to maximize the number of allocated traffic demands while minimizing the blocking ratio in a network with limited link capacity. The input parameters include the weighted graph $G(V, E)$, with the link $(i, j) \in E$, being characterized by $l_{i,j}$ and $c_{i,j}$, and the traffic matrix $T = [t_{s,d}]$, while the output parameters include the list of blocked traffic demands $B = [b_{s,d}]$ and the list of established paths $P = [\pi_{s,d}]$, with the path $\pi_{s,d}$ having the length $l(\pi_{s,d}) = \sum_{i,j} l_{i,j}$.

Furthermore, we assume that each path $\pi_{s,d}$ (also denoted as lightpath) computed using the CR approach is physically established using an optical channel with a specific wavelength, which is computed in this work using a first-fit heuristic [23]. In other words, a channel $k = (s, d)$, defined as $k = \{\pi_k, \lambda_k\} \in S$ has an associated path $\pi_k$ and wavelength $\lambda_k$, and belongs to the set of optical channels required to implement a logical full mesh topology $S = \{1, 2, \ldots, N(N-1)\}$. In the process of assigning wavelengths to the optical channels, which occurs during the routing process, it must be assured that all the optical channels that traverse the same link are assigned different wavelengths, as otherwise there would be interference between the channels. That means that there can be different channels using the same wavelength as long as there are no common links in their paths.

In this work, the CR problem is addressed through the following heuristic (CR heuristic):

(1) **Compute the shortest paths:**

- Run the Dijkstra's algorithm to find the shortest path between each source-destination node pair in the network $(\pi_{s,d})$, considering the total path length $l(\pi_{s,d})$ as the metric that defines that computation.

(2) **Order the traffic demands:**

- Apply a specific sorting strategy (e.g., shortest-first, longest-first, largest-first) to order traffic demands $t_{s,d}$. If the order is "shortest", the traffic demands are sorted by path length in ascending order, while for the "longest" order, the traffic demands are sorted by path length in descending order. Furthermore, if the order is "largest", the traffic demands are sorted by their value in descending order.

(3) **Route the demand, update link loads, and assign a wavelength:**

- For each traffic demand $t_{s,d}$, in accordance with the order established in Step 2, route it through $\pi_{s,d}$, updating the load (number of demands routed through the link) of each link in $\pi_{s,d}$, and assign a wavelength $\lambda_k$ to that optical channel (a wavelength being represented by an integer between 1 and $N_{ch}$).

(4) **Blocking:**

- If, in Step 3, a link (or more than one) in $\pi_{s,d}$ does not have enough residual capacity (which is defined as the difference between the link capacity and its load), or if a wavelength that fits all links of the path does not exist (respecting the principle that two optical channels with the same wavelength cannot exist on the same link), then the traffic demand $t_{s,d}$ is blocked.

(5) **Remove links and determine alternative shortest paths:**
   - After routing each traffic demand, remove all the links that have residual capacity zero from the weighted graph.
   - With the updated topology, determine new shortest paths, as in Step 1, so that alternative paths are found for the remaining traffic demands.
   - Go to Step 3 to route the next traffic demand.

To compute the total network capacity, one can apply the concepts of channel capacity introduced in Section 2, which can be written as [14]

$$C_{net} = \sum_{k \in S} C_{ch,k} \tag{13}$$

where $C_{ch,k}$ is the capacity of channel $k$, which, according to (11) and (13), becomes:

$$C_{ch,k} = 2R_s \log_2(1 + SNR_k) \tag{14}$$

with $SNR_k$ being the $SNR$ of channel $k$. The $SNR_k$ can be readily evaluated using (4), assuming that the optical nodes (ROADMs) are ideal and, as a result, do not affect the calculations. In this context, the number of spans for optical channel $k$ is denoted as $n_{s,k} = \lfloor L_k / L_s \rfloor$, with $L_k$ representing the length of the path $\pi_k$. To avoid calculating the $SNR_k$ and reduce the computation time, we can take advantage of the analysis undertaken in Section 2 and use the optical reach to obtain the channel's capacities. By knowing the lengths of the different paths and utilizing the data from Table 1, we can obtain the capacities of the different channels for a span length of 80 km and for the two symbol rate values (64 Gbaud and 128 Gbaud) from Table 2. These capacities are referred to as Shannon capacities because the reach values are obtained using the Shannon theory.

An additional important metric for network analysis is the network-wide average channel capacity, defined as [24]

$$\overline{C}_{ch} = \sum_{k \in S} C_{ch,k} / \sum_{k \in S} \gamma_k \tag{15}$$

where $\gamma_k$ denotes the expected utilization ratio of channel $k$. For the sake of simplicity, it is assumed that $\gamma_k = 1$ for all channels. As a result, the sum in the denominator of (15) equals the total number of paths in the network, which, for a full-mesh logical topology, amounts to $N(N-1)$. With this simplification, the network capacity for a full-mesh logical topology is reduced to

$$C_{net} = \overline{C}_{ch} \times N(N-1) \times (1 - \overline{B}) \tag{16}$$

where $\overline{B}$ is the average blocking ratio obtained as

$$\overline{B} = \sum_{k \in S} b_k / (N(N-1)) \cdot \tag{17}$$

## 5. Unconstrained Routing and Fiber Assignment

Optical backbone networks are typically designed to avoid blocking traffic demands. Blocking occurs when there is insufficient capacity to accommodate all the incoming traffic demands at a particular node or link. In the previous analysis, blocking occurred due to the limited number of optical channels and their corresponding wavelengths on each link. This limitation arises, namely, from bandwidth constraints of the optical amplifiers, which, in this work, are assumed to be operating in the C-band. To address the blocking problem

in optical backbone networks, one can utilize optical amplifiers that operate in other bands different from the C-band, such as the L-band and the S-band. Nevertheless, this solution has some drawbacks: one can refer, for example, to the technical difficulties associated with building optical amplifiers to operate in the S-band and the need to add band multiplexers/ demultiplexers to separate the different bands for individual amplification, which can significantly increase the transmission losses.

A more straightforward solution for increasing the overall capacity of an optical backbone network is to add more optical fibers per link. However, this can be a costly and complex solution, particularly when extensive upgrades are required. Nonetheless, in common scenarios where network operators own dark fibers, lighting additional fibers emerges as a viable and cost-effective solution. This study will explore this approach as a means of overcoming blocking. To achieve this objective, a fiber-assignment heuristic, designated FA heuristic, will be proposed. The input parameters of this heuristic are also a weighted graph $G(V, E)$, as in the CR heuristic , but now with $c_{i,j} = \infty$ (meaning that there is no constraint relative to the number of optical channels in a given link), the traffic matrix $T = [t_{s,d}]$ and the maximum number of available optical channels per fiber $N_{max} = N_{ch}$. On the other hand, the output parameters comprise the list of established paths $P = [\pi_{s,d}]$ with the path $\pi_{s,d}$ having the length $l(\pi_{s,d})$, as in the CR heuristic, and an $N \times N$ matrix with the number of optical fibers per link, $NF = \left[ nf_{i,j} \right]$, where $nf_{i,j}$ is the number of optical fibers in the link $(i, j)$. The first part of the FA heuristic is equivalent to Steps 1–3 of the CR heuristic, but now using an unconstrained routing strategy, which permits obtaining the list P, and an $N \times N$ matrix with the wavelengths in each link, $W = \left[ w_{i,j} \right]$, where $w_{i,j}$ is the list of all the wavelengths $\lambda_k$ present in the link $(i, j)$, $w_{i,j} = [\lambda_k]$. Subsequently, the next steps of the heuristic are the following:

(4)  **Assign fibers when there is no traffic in a link:**

- If there is no traffic in that link but the link does exist in the network's physical topology, set $nf_{i,j} = 1$

(5)  **Assign fibers when there is traffic in a link:**

- Set $nf_{i,j} = \max(num\_rep\_w_{i,j})$, where $num\_rep\_w_{i,j}$ is the number of repeated wavelengths in $w_{i,j}$, $\forall (i, j) \in E$

Note that in this context, where unconstrained routing is being done, the number of wavelengths in each link does not have a limit, so the value attributed to a given $\lambda_k$ is any natural number (and not bounded by $N_{max}$, as in the CR heuristic ). To determine the number of fibers needed in each link, the maximum number of "repeated wavelengths" in that link needs to be determined. A wavelength is considered a "repeated wavelength" when its value modulo $N_{max}$ (the modulo operation referring to the remainder of a division) is equal to that of another wavelength also present in that link. For instance, if $N_{max}$ is 75, then wavelengths 1 and 76 are "repeated" because 76 modulo 75 equals 1. This implies that both wavelengths would occupy the same channel in a link; hence they are "repeated". This concept is crucial in determining the number of fibers needed for a link, ensuring that each "repeated" wavelength has its own fiber. Finding the maximum count of "repeated wavelengths" will ensure that there are enough fibers to accommodate all the wavelengths, thus assuring that there are no channels with the same wavelength on the same fiber.

By knowing the length $l(\pi_{s,d})$ of all the paths belonging to P, it is possible to compute the capacity of the optical channel corresponding to those paths using the values of the reach given in Table 2 and consequently computing the average channel capacity using (15) and total network capacity using (16) with $\overline{B} = 0$. To assess the network performance in the present scenario, it is also necessary to account for the network cost. For simplification purposes, we assume that the transponder cost can be neglected in comparison with the

fiber cost, which seems to be a reasonable assumption for optical backbone networks [25]. In this case, the network cost is given as

$$\Lambda_{net} = \sum_{i,j} l_{i,j} \times nf_{i,j}. \tag{18}$$

## 6. Results and Discussion

To investigate the dependence of network capacity on network parameters, five sets, each comprising 200 graphs, were obtained using the modified Waxman model described in Section 3, with the number of nodes varying from 20 to 60 in increments of 10. All the graphs were generated assuming a bi-dimensional plane with dimension $1000 \times 1000$ km and Waxman parameters $\alpha = \beta = 0.4$, as well as an average node degree varying randomly from 2 to 4. These sets of random networks were used in both routing scenarios described previously, i.e., constrained routing (Section 4) and unconstrained routing with fiber assignment (Section 5).

In the first scenario, the routing was performed considering a full-mesh logical topology described by the traffic profile (12), using the CR heuristic and the shortest-first sorting strategy. The study was undertaken for two symbol rate values, 64 Gbaud and 128 Gbaud, considering the optical reach values provided in Table 2 and one fiber pair per link, with each fiber being used in a communication direction. Furthermore, the number of optical channels per fiber was limited to 75 for 64 Gbaud and 37 for 128 Gbaud. This limitation arises from the fixed bandwidth of 4800 GHz in optical amplifiers and by considering a channel spacing of 64 GHz and 128 GHz for 64 Gbaud and 128 Gbaud transmissions, respectively. Note that the equality between channel spacing and symbol rate results from the Nyquist limit assumption, as explained in Section 2.

The values of the computed total network capacity are depicted in Figure 3 using boxplots. A boxplot is a way of illustrating the statistical distribution of a data set and includes the median, the interquartile range, and both the minimum and maximum values of the set. The boxplots in Figure 3 also show outliers, represented as small circles, to describe data samples that differ significantly from the rest of the data set.

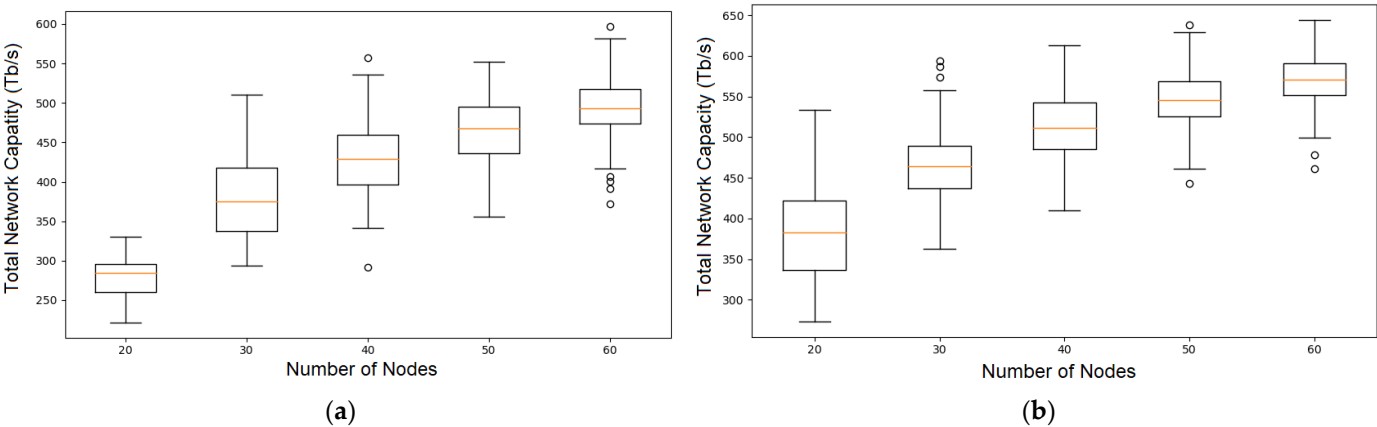

**(a)**                                                       **(b)**

**Figure 3.** Total network capacity in the case of constrained routing: (**a**) 75 channels, 64 Gbaud; (**b**) 37 channels, 128 Gbaud.

It can be seen from Figure 3 that the network capacity tends to grow as the number of nodes increases, although the rate of growth decreases for higher node counts. This increase in the network capacity is expected according to the relation between the number of nodes and the network capacity described in (16). The fact that the capacity growth tends to be slower as the number of nodes increases indicates that the blocking of traffic demands must also be increasing with the number of nodes. Figure 4, which depicts both the number of blocked traffic demands and the blocking probability, which is obtained by dividing the number of blocked traffic demands by $N \times (N-1)$, shows that is indeed the case. This

increase in blocking occurs because networks with more nodes experience a higher volume
of traffic demands (as described by (12)). Consequently, the conditions for blocking (see
Section 4) intensify as the links become increasingly saturated with traffic demands.

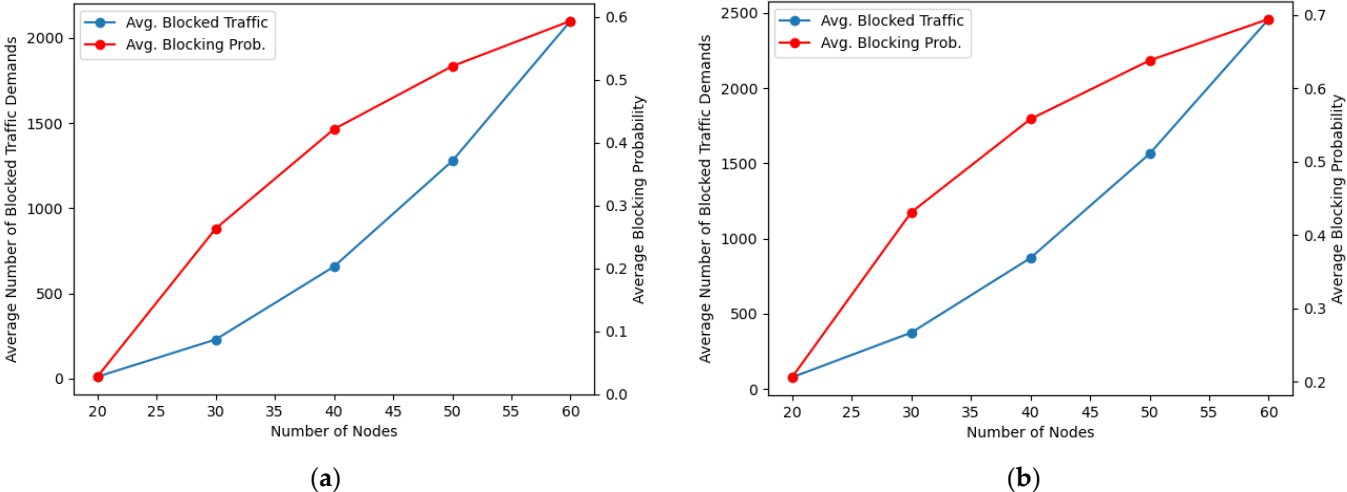

**Figure 4.** Average number of traffic demands and average blocking probability as a function of the
number of nodes (average across each set). (**a**) 75 channels, 64 Gbaud; (**b**) 37 channels, 128 Gbaud.

When comparing the transmission at 64 Gbaud (Figure 3a) and at 128 Gbaud (Figure 3b),
we can see that using a symbol rate of 128 Gbaud makes it possible to achieve a higher
total network capacity in comparison with the 64 Gbaud case, although the extent of the
improvement tends to decrease as the number of nodes increases. Comparing the median
capacity values between the sets of generated graphs, transmission at 128 Gbaud results in
an improvement over the transmission at 64 Gbaud of approximately: 34%, 24%, 19%, 17%,
and 16%, for the respective sets of graphs, listed in ascending order of number of nodes.
The average improvement across all sets is around 22%. The decrease in performance
improvement verified in networks with more nodes can be explained by the slight increase
in the blocking probability, which, for example, for the case of 60 nodes, rises from 0.6 to
0.7 as the symbol rates go from 64 Gbaud to 128 Gbaud, as can be seen in Figure 4. The
improvement in the networks' capacity, coupled with the simultaneous reduction in the
number of wavelengths, which are halved, represents an important advantage in utilizing
128 Gbaud compared to 64 Gbaud.

The previous analysis deals with the constrained routing of traffic demands due to
the limited number of optical channels per link. As can be seen, this leads to blocking,
which increases with the size of the network, as shown in Figure 4. To address the blocking
problem, one can enhance the link capacity by adding more optical fibers, following the
strategy outlined in the FA heuristic. As a result, the network achieves an unconstrained
total capacity, determined only by the load of the traffic demands, without any imposed
constraint. This capacity is shown in Figure 5, which also uses boxplots. As in the first
scenario, the traffic profile described in (12) was also considered, as well as the shortest-first
sorting strategy. The values of the maximum number of optical channels per fiber ($N_{max}$)
were set to 75 for transmission at 64 Gbaud and 37 for transmission at 128 Gbaud. As
described in the FA heuristic, there is no limit set on the addition of fibers, so enough fibers
are added to eliminate the blocking completely.

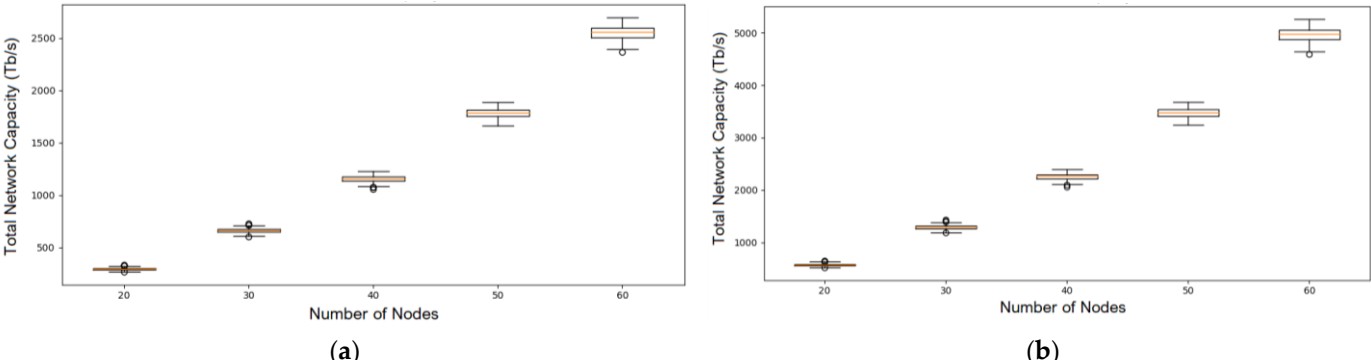

**Figure 5.** Total network capacity in the case of unconstrained routing: (**a**) 64 Gbaud; (**b**) 128 Gbaud.

The first conclusion we can draw from Figure 5 is that the total capacity increases approximately in a quadratic manner with the number of nodes ($\sim N^2$). Another noteworthy aspect is the huge capacities achieved in this scenario, which corresponds to about 2.5 Pbit/s for 60-node networks and a symbol rate of 64 Gbaud (see Figure 5a). It can also be referred to that the total network capacity median values for a 30-node network (~660 Tbit/s) are similar to the values reported in Figure 9 of [14] for the 30-node CONUS topology generated using the Erdős–Rényi model. As expected, Figure 5b shows a twofold increase in the total capacity when the symbol rate is set at 128 Gbaud.

According to what is expected, the significant increase in capacity comes at the cost of a substantial rise in network cost, which translates into an increase in the optical fiber length to be deployed. Figure 6 shows the total fiber cost, expressed in terms of the total fiber length, as a function of the number of nodes. This figure shows a law of variation of the cost as a function of the number of nodes similar to the one of the capacity referred to above. A prominent conclusion we can draw from Figure 6 is that when the symbol rate increases from 64 Gbaud to 128 Gbaud, the total fiber cost increases by about 51%, while the total network capacity value doubles, as seen previously.

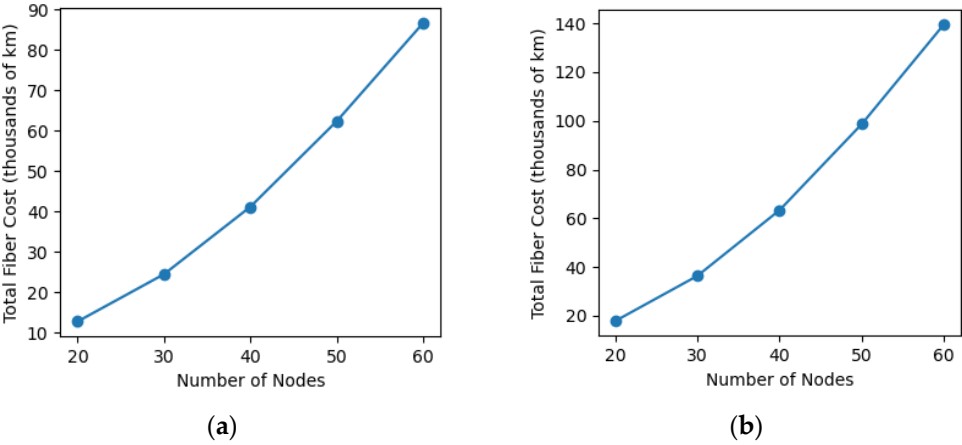

**Figure 6.** Total fiber length (average across each set). (**a**) 64 Gbaud; (**b**) 128 Gbaud.

## 7. Conclusions

In this paper, the problem of assessing the impact of topological and physical impairments on the capacity of optical backbone networks was investigated.

The capacity was defined using Shannon's theory, and the impact of the physical layer was studied using the optical reach, which was computed, considering both linear and nonlinear noise terms, for two symbol rate values: 64 Gbaud and 128 Gbaud. To explore the influence of topological characteristics, we used a modified Waxman model to generate random networks that mimic real optical backbone networks, ensuring edge connectivity greater than or equal to 2.

The paper also proposed a constrained routing and wavelength assignment algorithm to deal with the fact that the number of optical channels/wavelengths per link is limited, which inevitably results in traffic blocking as the number of demands increases. Given that traffic blocking is not acceptable in optical backbone networks, we also devised a strategy to overcome it by adding more optical fibers per link, albeit at the expense of increasing the network cost.

The total network capacity was evaluated for a set of generated random networks considering a full-mesh logical topology. The results showed that although the capacity increases with the number of nodes, the rate of increase tends to diminish due to the rising of the blocking ratio. By moving from a symbol rate of 64 Gbaud to 128 Gbaud, one observes an improvement in median total capacity of about 34% for $N = 20$ and 16% for $N = 60$. The reduction in improvement is also explained by the rising of the blocking ratio. With proper fiber assignment, one can see a substantial increase in the total capacity. For a network with $N = 60$, median values of about 2.5 Pbit/s can be achieved for a symbol rate of 64 Gbaud and about 5 Pbit/s for a symbol rate of 128 Gbaud. Remarkably, this duplication in the total network capacity is achieved by an increase in the total fiber length of only about 51%.

**Author Contributions:** Conceptualization, A.F. and J.P.; methodology, A.F. and J.P.; software, A.F.; validation, A.F.; formal analysis, A.F. and J.P.; investigation, A.F. and J.P.; writing—original draft preparation, A.F. and J.P.; writing—review and editing, A.F. and J.P.; visualization, A.F.; supervision, J.P. All authors have read and agreed to the published version of the manuscript.

**Funding:** This research received no external funding.

**Data Availability Statement:** Data are contained within the article.

**Conflicts of Interest:** The authors declare no conflicts of interest.

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
