# Peer review of "Investigating the Impact of Topology and Physical Impairments on the Capacity of an Optical Backbone Network"

_photonics, doi:10.3390/photonics11040342_

Round 1

Reviewer 1 Report

Comments and Suggestions for Authors

The paper focuses on studying the impact of topology and physical impairments on the capacity of optical backbone networks. The main conclusion is that capacity does not increase linearly with network size due to increase in blocking. I do not understand clearly this effect: does increasing the number of nodes increase the path lengths or is it due to a larger number of hops in the paths? I would appreciate to further elaborate on this point.

Indeed, if traffic between nodes is totally uniform, probably static “off-line” routing and spectrum assignment would work better.

My main concern with the manuscript is about the traffic model. In line 274 authors say that “the traffic profile is uniform and equal among all node pairs“. I do not see how table 2 impacts this uniformity. When referring to traffic is it bit/s? Please, clarify this because I detect some confusion with these capacity issues (did authors considered capacity as a function of length??).

In fig. 4 I do not see why convexity of the curves change. Is it any explanation?

Authors talk about blocking ratios of 0.6 and 0.7 (e.g. line 452). I would prefer working with blocking probabilities below 10% which are typical in these optical transport network studies. These so high blocking values are not realistic.

Differences between constrained and unconstrained routing should be better explained. Is it the only difference an increase in available fibres?

 Probably some references more recent than 8,9, 19 can be found for maximum transmission reach.

Reviewer 2 Report

Comments and Suggestions for Authors

A method to evaluate the network capacity that relies on the optical reach to account for physical layer aspects, in conjunction with constrained routing techniques for traffic routing is proposed. Furthermore, a fiber assignment algorithm was designed to deal with the blocking caused by the limitation on the number of wavelengths per fiber. To some point, the researches have scientific research significance and certain innovation. The paper is organized properly and the work does provide useful insight for the evaluation of the impact of topology and physical impairments on network capacity in optical backbone networks.

Some questions in the following:

1. From line 245 to 248 proposed In the subsequent steps, nodes are inter-connected within each region and then across different regions according to the Waxman probability,...”. It is recommended to indicate in Figure 1 which part is the inter-connected within each region and which part is across different regions. In addition, it is recommended to increase the character size in the figures.

 2. From line 432 to 433, “Furthermore, the number of optical channels per fiber was limited to 75 for 64 Gbaud and 37 for 128 Gbaud”, what is the basis for this statement? Can the author give theoretical support or literature reference?

 3. In line 42, “WDM (Wavelength Division Multiplexing)”, write the abbreviation first, then the full name; but in line 77, “Gaussian noise (GN)”, write the full name first, then the abbreviation. It is recommended that author abbreviations be kept in a uniform format.

 4. In line 175, b2, the fiber dispersion in ps-2km-1, but in Table 1, the fiber dispersion parameter is -21.7 ps2km-1. The units of fiber dispersion are different.

Comments on the Quality of English Language

Minor editing of English language required.

Round 2

Reviewer 1 Report

Comments and Suggestions for Authors

Authors have apropriately addressed my concerns so the manuscript can be published in its present form.